# Facade-Based Bicelles as a New Tool for Production of Active Membrane Proteins in a Cell-Free System

**DOI:** 10.3390/ijms241914864

**Published:** 2023-10-03

**Authors:** Marina V. Goncharuk, Ekaterina V. Vasileva, Egor A. Ananiev, Andrey Y. Gorokhovatsky, Eduard V. Bocharov, Konstantin S. Mineev, Sergey A. Goncharuk

**Affiliations:** 1Shemyakin-Ovchinnikov Institute of Bioorganic Chemistry of the Russian Academy of Sciences, Moscow 117997, Russia; m.s.goncharuk@gmail.com (M.V.G.); angor@ibch.ru (A.Y.G.); edvbon@mail.ru (E.V.B.);; 2Moscow Institute of Physics and Technology, Dolgoprudny 141701, Russia

**Keywords:** cell-free, CFPS, membrane protein, protein production, synthesis of an active protein, bicelle, Facade, nanodisc, lipid, detergent

## Abstract

Integral membrane proteins are important components of a cell. Their structural and functional studies require production of milligram amounts of proteins, which nowadays is not a routine process. Cell-free protein synthesis is a prospective approach to resolve this task. However, there are few known membrane mimetics that can be used to synthesize active membrane proteins in high amounts. Here, we present the application of commercially available “Facade” detergents for the production of active rhodopsin. We show that the yield of active protein in lipid bicelles containing Facade-EM, Facade-TEM, and Facade-EPC is several times higher than in the case of conventional bicelles with CHAPS and DHPC and is comparable to the yield in the presence of lipid-protein nanodiscs. Moreover, the effects of the lipid-to-detergent ratio, concentration of detergent in the feeding mixture, and lipid composition of the bicelles on the total, soluble, and active protein yields are discussed. We show that Facade-based bicelles represent a prospective membrane mimetic, available for the production of membrane proteins in a cell-free system.

## 1. Introduction

Membrane proteins (MPs) as a fundamental molecular component of a cell mediate many key processes: signal transduction, component transport, enzymatic reactions, communications between the cells, etc. Malfunction of MPs leads to a wide spectrum of socially important diseases including cancer, neurodegenerative, and cardiovascular pathologies [1,2,3,4,5]. They represent the majority of current drug targets and attract significant interest from pharma and biotech [6,7,8]. Nevertheless, despite the significant role of MPs, this class of proteins remains poorly understood. One of the main problems is their production in the active state for functional and structural studies in milligram quantities.

To date, there are many tools for protein production in cell cultures, but obtaining a large amount of active MP remains a non-trivial task [9,10]. Due to their hydrophobicity, MPs tend to form insoluble aggregates or to insert into the cell membrane, causing host lysis or low protein yields. Refolding of MPs from the aggregates or unfolded states using either harsh detergents or organic solvents is also possible [11,12,13] but requires extensive condition screening. Achievements of the last decades have made cell-free systems (CFs) a good alternative for this aim [14,15]. By adding membrane-like components to CF reaction mixtures, one could achieve co-translational folding of the desired MP [16,17,18], and there are many successful examples of MP production using CFs [19,20,21,22,23,24,25,26]. On the other hand, there is no general approach to synthesizing an arbitrary MP in CFs. The main obstacle is the selection of membrane mimetics that can support the native folding of an MP, self-assemble into particles small enough to form stable suspensions, and do not interact with CF components like enzymes and ribosomes. Detergent micelles, bicelles, and nanodiscs were successfully used for this purpose. Lipid-protein nanodiscs (LPNs) have shown a significant superiority over other mimetics [22]; however, they have several drawbacks, including relatively high cost, which can be reduced by the in-home synthesis of belt proteins, and the hindered exchange of matter between the particles [27]. While there are examples, with protein complexes being co-translationally incorporated into LPNs [28], further possible manipulations with protein samples obtained in LPNs are limited. Therefore, the search for other “open” membrane-mimicking media remains an urgent task.

Facades are a relatively new class of soft amphiphilic detergents [29,30] (Figure 1B). The molecules belong to the class of facial amphiphiles and are believed to have two sides with distinct physical properties. Whereas their nonpolar face is formed by a cholesterol-like core of cholic acid, their polar sides are made by modifying hydroxyl groups either by carbohydrate moieties (Facade-EM (F-EM), Facade-TEM (F-TEM), and Facade-TEG (F-TEG)) or by phosphocholine groups (Facade-EPC (F-EPC)). Facade-TFA1 (F-TFA) represents two covalently linked cholic acid cores modified by glycans. There are several works demonstrating the successful application of Facades to study membrane proteins [31,32,33,34,35]. Earlier, we showed that F-EM and F-EPC, mixed with lipids, can form isotropic bicelles with no mixing of lipid and detergent, and may be used for NMR studies of membrane proteins [36,37]. Thus, these molecules are promising options that could be tested for membrane protein production in cell-free systems.

Here we used a continuous-exchange cell-free protein system based on *E. coli* S30 extract to investigate Facade-based bicelles as a membrane component for CF synthesis. We studied the influence of several parameters, including the size and composition of bicelles and the concentration of detergent in reaction and feeding mixtures on protein yield and activity. We show that Facades bicelles are a prospective membrane mimetic for MP production in CFs.

## 2. Results

In order to test membrane protein synthesis in a continuous-exchange cell-free system based on *E. coli* S30 extract, we used bacteriorhodopsin from *Exiguobacterium sibiricum* (ESR) [38,39]. In a sense, rhodopsins are unique objects, since they are the only transmembrane proteins, the activity of which could be easily assayed directly inside a CF vial by following their visible light absorption. In our experiments below, we synthesized ESR in the presence of retinal and different lipid and detergent additives and quantified three parameters: (1) the activity of ESR by UV–Vis spectroscopy and overall yields of (2) total and (3) soluble protein using Western blot analysis. CF synthesis in the absence of any membrane mimetics was used as a control. Here and below, ‘soluble protein’ stands for the protein that remains in solution after centrifugation, most likely embedded into the particles of membrane mimetic environment, but may not be active, whereas ‘active protein’ means a soluble and properly folded ESR. Such an analysis allows simultaneous assessment of both the ability of membrane mimetics to provide proper protein folding and their influence on the activity of translation machinery in the CF reaction mixture. A scheme of the experiment is shown in Figure 1A.

### 2.1. Facade-Based Bicelles Reveal Prospects as a Membrane Mimetic Additive for Cell-Free Systems

To obtain a first impression of Facade utility in a CF synthesis, we performed a wide screening of various membrane mimetics, including micelles, bicelles, lipid-protein nanodiscs (LPNs), and liposomes. In addition to five commercially available Facades, a classic MP solubilization detergent, n-Dodecyl-β-D-Maltoside (DDM), and commonly used bicelle rim-forming detergents 1,2-dihexanoyl-sn-glycero-3-phosphocholine (DHPC) and 3-[(3-cholamidopropyl)dimethylammonio]-1-propanesulfonate (CHAPS) were explored (Appendix A). The bicelle rim-forming detergents were used separately and together with 1,2-dimyristoyl-sn-glycero-3-phosphocholine (DMPC) to check the effect of bicelle formation. The results are provided in Figure 2.

First, the total protein yield in the presence of micelles of conventional detergents (DDM, CHAPS, and DHPC), as well as Facade-EPC and DMPC liposomes, was significantly decreased (Figure 2), while with other Facades it was just 1.5–2 times lower compared to the control or even equal in the case of F-TFA (~0.8 mg per mL of the reaction mixture (RM)). This is in agreement with previous findings that reveal the toxic effects of conventional detergents [40]. A significant yield of soluble ESR was observed for three detergents: F-TFA, F-EM, and F-TEM, starting from 30% compared to the total yield of the control sample. Additionally, F-TEM showed the comparable yield of soluble protein with respect to total yield. At the same time, no ESR activity was detected for all the detergents.

A distinct behavior was observed for DMPC bicelles. A certain protein activity was detected for all the rim-forming detergents, excluding CHAPS and F-TEG. Moreover, ESR activity in the presence of F-EM and F-TEM bicelles was comparable to LPNs, while somewhat lower. The yields of soluble ESR for F-EM and F-TEM bicelles and LPNs were comparable with the total yields, and they were close to control.

Thus, here we found that Facade-based bicelles, especially F-EM and F-TEM, can be successfully applied as membrane mimetics for co-translational folding during the cell-free synthesis of membrane proteins. They demonstrate a much lower inhibiting effect on the translation machinery than other detergent-based mimetics and allow the production of soluble and active ESR.

### 2.2. Size of Bicelles Is an Important Parameter of a Cell-Free Reaction

To continue, we selected one of the Facade detergents, F-EM, and further optimized various parameters of the CF reaction. F-EM was used for economic reasons—it is substantially cheaper than F-TEM, which demonstrated comparable efficiency in the initial screening. Besides this, the size, shape, and phase transitions in F-EM-based bicelles are well established, unlike the other compounds [36,37]. One of the important parameters governing bicelle behavior is the lipid-to-detergent ratio (q) [41,42,43,44,45,46]. A lower q corresponds to a smaller size of bicelle and, hence, a smaller size of the lipid bilayer. Since these features may affect protein stability and activity, we decided to analyze the effect of bicelle size on protein synthesis. Total, soluble, and active yields of ESR at different q were normalized to the corresponding expression levels of total, soluble, and active protein in DMPC:F-EM at q = 0.5 (the same data point as in the first experiment—DMPC:F-EM bars in Figure 2).

As one can easily notice, the yield of active ESR was significantly increased in the range of 1 ≤ q ≤ 3 (Figure 3A). The optimum was at q = 2, while there was no observed statistically significant difference between q = 1, 1.5, 2, and 3, having radii of ~3.3, 3.9, 4.5, and 5.7 nm, respectively [36]. The yield of active ESR at q = 2 was up to 1.8 times higher compared to q = 0.5. Interestingly, the total and soluble protein yields did not change significantly up to q = 3. At the same time, at higher q values (q = 5 and q = 10), the protein synthesis decreased to a negligible level, probably due to changes in bicelle morphology.

A similar effect was observed for the bicelles containing CHAPS or DHPC (Figure 3B,C). The yields of active protein increased by factors of ~5 and ~2, respectively. For DMPC:DHPC, the yields of total and soluble protein increased by up to 2–3 times. Surprisingly, whereas for DMPC:CHAPS bicelles the total yield increased by about two times, the level of soluble protein increased by up to one order of magnitude. Thus, after optimization of q, the yields of soluble and active ESR for DHPC and CHAPS bicelles became comparable with each other but were still significantly lower than for F-EM (Figure 3D).

### 2.3. The Detergent Concentration in the Feeding Mixture Has a Limited Influence on the Yield of Active Protein

Another parameter that can affect cell-free reactions is the detergent concentration in the feeding mixture (FM). Several detergent concentrations in the reaction mixture were used above to make bicelles with different q. At the same time, the concentration in FM was fixed to critical micelle concentration (CMC) to prevent detergent dilution. However, in the context of bicelles, the free detergent concentration can differ markedly from the CMC and depends on q [41,42,47]. Moreover, the exchange between the RM and FM in the course of translation may result in a variation in the effective q ratio in RM, which in turn may affect the protein yields as described above. Thus, fixing the lipid concentration at 15 mM DMPC, we compared the influence of detergent concentration in the FM on protein yields in the presence of DMPC:F-EM bicelles at two concentrations in the RM: 30 mM, corresponding to q = 0.5, and 7.5 mM, corresponding to q = 2.

In both cases, the behavior was very similar (Figure 4). The yield of total and soluble proteins remained almost unchanged. A significant effect on the active protein yield was detected when concentrations of detergent in RM and FM were equal. The level of active protein was increased by up to 1.4 times at q = 0.5 and up to 1.2 times at q = 2.

### 2.4. Facade-EM Is a Versatile Rim-Forming Detergent for Different Lipids

We have shown that Facades can be successfully used in the context of DMPC bicelles in CF synthesis, but the following question arises: “Could Facades be used with other types of lipids?”. To answer this question, we screened lipids with different characteristics: DMPG as a lipid with a negative charge, DLPC with shorter tails, POPC, and DOPC with longer and unsaturated tails.

The addition of a negative charge to the bicelles resulted in an increase in the yield of active protein by up to 30% (Figure 5A). Surprisingly, this effect does not depend on the amount of DMPG in the range of 10–100%, although for bicelles containing 100% DMPG, the optimum q was equal to 1 (Appendix A). Possibly, the negative charge of the membrane is a more native environment for the ESR. Many works revealed that negatively charged lipids may affect the insertion of transmembrane proteins into the cell membrane and even alter their fold and transmembrane topology [48]. Alternatively, this effect could be associated not with an increase in the amount of an active protein, but rather with a change in the extinction coefficient in the presence of a negatively charged lipid or with a shift of equilibrium to another state of rhodopsin in the photocycle [49,50]. The DMPG dose-independent effect can be explained by the specific binding of the lipid to the ESR, for example, if the protein binds only a few molecules of charged lipid with high affinity and thus stabilizes the structure.

In the presence of bicelles containing POPC or DOPC, protein was produced as a precipitate (Figure 5A). Moreover, the total protein yield was decreased for POPC bicelles. However, it is noteworthy that the precipitates were bright purple, indicating the active state of the ESR (Appendix A). The same behavior took place in the case of precipitates for DMPC:F-EM for q in the range: 2 < q < 5 (Appendix A). We assumed that the concentration of F-EM is not sufficient to form bicelles. Thus, the protein is transferred to liposomes or large lamella and precipitates in complex with lipids at high q values. The lower propensity of POPC and DOPC to form bicelles at high q agrees with the observation that POPC bicelles are generally more susceptible to temperature-induced growth and form opalescent solutions even at moderately low q values [41,51]. To test this hypothesis, we increased the concentration of detergent and tested bicelles with q = 1 and q = 0.5 for both POPC and DOPC. Indeed, with a decrease in q, a significant increase in soluble and active protein was observed, while the maximal yields were lower than those obtained for DMPC:F-EM and q = 2 (Figure 5B, Appendix A).

Thus, we showed that Facade-EM can be used in combination with different types of lipids to produce an adequate membrane-like environment during CF synthesis.

### 2.5. Facade-EM Bicelles Are the Most Promising Additives for the Co-Translational Folding of Membrane Proteins

Based on the data obtained for DMPC:F-EM bicelles, we decided to test other Facades at the conditions revealed for F-EM to be more optimal and to compare these data with LPN and control (CF reaction without any membrane mimetics) samples. We used 15 mM of DMPC, 2 mM of Facades in FM (in excess with respect to the CMC), and 7.5 mM of Facades in RM (q = 2), except F-TEG (15 mM, q = 1), since it was not possible to obtain a clear solution of bicelles with this detergent at room temperature.

No protein expression was detected in the presence of DMPC:F-TEG (Figure 6), while in the first experiment at q = 0.5, we observed protein expression, but without soluble and active protein (Figure 2). This could be explained by the fact that F-TEG is not able to form bicelles at high q—highly opalescent solutions are formed at q above 0.5.

Other DMPC:Facade bicelles revealed similar total yields of ESR, which were comparable to the control (Figure 6). However, in the presence of F-TFA, the yields of soluble and active ESR were significantly lower. It is noteworthy that at q = 0.5, the amount of soluble protein was comparable with the total yield (Figure 2). We believe that a higher detergent concentration is required for F-TFA and F-TEG to produce stable and soluble bicelles.

F-EM, F-TEM, and F-EPC demonstrated high yields of active ESR and they were comparable with LPN (Figure 6). In general, the overall best result was detected for F-EM (~0.6 mg of active protein per mL of the RM). The expression level of active ESR for F-EM bicelles was similar to total yield and comparable with the yield of active protein in LPN. Notably, the obtained result is also comparable to the yields reported previously for ESR in LPNs of different composition (0.6–0.9 mg/mL, using phosphoenol pyruvate/pyruvate kinase system [22]). It is probable that for F-TEM and F-EPC, additional optimization is necessary to find the best conditions.

## 3. Discussion

The production of natively folded and functional membrane proteins is one of the key problems of modern structural biology. Cell-free protein synthesis is a prospective approach to resolve this problem. However, the membrane or specified membrane mimetic is required for proper folding. There are many examples of detergent, bicelles, liposomes, and lipid-protein nanodiscs used to produce MPs. The latter show significant advantages with respect to others [22]. They do not inhibit the translation machinery and perform extremely well in stabilizing protein samples [52,53]. At the same time, LPNs have some bottlenecks. First, LPNs almost do not exchange components. The exchange of lipids between LPNs has rates comparable to those observed in small lipid vesicles and three to four orders of magnitude lower than the ones in micelles or lipodiscs, and most likely follow the monomer diffusion mechanism [27]. While one can co-translationally assemble relatively large protein complexes in LPNs [28], mixing of proteins from two different LPN samples is hardly possible. This could complicate the study of protein–protein interactions in the membrane. Second, LPNs comprise the rim-forming protein—MSP or its analogs [54]. It can interfere with the study of the target protein, for example, by interacting with it. These problems can be solved by using another type of nanodisc—SMALPs, where MSP is replaced by a styrene maleic anhydride polymer, but to date, there is no systematic analysis of this mimetic in the context of CF synthesis and only several examples of its application have been published [55,56]. Thus, the search for new membrane agents is an ongoing task.

Here, we show that bicelles based on commercially available Facade detergents are an excellent membrane mimetic for cell-free protein synthesis. Except for F-TEG, they do not disrupt the transcription/translation systems and, as a result, do not affect total protein yields. In contrast to classical bicelles based on CHAPS or DHPC, they show several times’ more efficient incorporation of active ESR. The yield of active protein in F-EM bicelles was comparable to LPNs. At the same time, bicelles have several advantages in contrast to LPNs. They can easily exchange components, thus allowing the study of protein oligomerization kinetics and other processes in dynamics [57,58]. Additional components, like specific lipids or drugs, can easily be incorporated into bicelles. Moreover, bicelles can be softly and fully exchanged to another membrane mimetic: micelles, bicelles with another composition, LPNs, or liposomes, for example, by immobilized affinity chromatography or dialysis, although this option is also available for LPNs through the stage of bicelle formation [59].

Another important observation here is the influence of lipid-to-detergent ratio (q) on protein yields. As is known, low q values correspond to a smaller size of lipid bilayer; therefore, the protein can interact with the detergent, which leads to protein instability and its partial unfolding. On the other hand, at high q values, bicelles cannot be formed and the protein precipitates within the liposomes or lamellae. While the optimum was observed for all investigated bicelles, the best q values were different and depended on bicelle composition (Figure 3, Appendix A). Thus, to produce the target membrane protein in the CF system using bicelles, additional screening to find the optimal q is required.

The increase of concentration of detergent in the FM to the same in RM also increases the yield of the active protein (Figure 4). However, this requires a great amount of detergent, since the ratio of FM to RM is usually in the range of 4–30 [14,60]. The high cost of Facades can make the protein sample very expensive. On the other hand, the decrease in protein yield at CMC in FM was only about 20%. Whereas the Facades, and especially F-EM, are characterized by a low CMC value (~0.1 mM), it may be advantageous to use this concentration for protein production in CF.

On the other hand, we would like to point out that all the results here are based on the example of bacteriorhodopsin, which is a convenient model but is known to be a well-behaved protein and cannot be considered representative of all MP classes. Thus, while the effect of Facade detergents on translation machinery is independent of the type of synthesized MP, their ability to support the protein folding and stability needs to be verified in every particular case, because various proteins are known to require the individual membrane-like environment [16,23,61].

Finally, in some experiments, like those with high q values or with DOPC or POPC lipids, the ESR expressed normally, but the yield of soluble protein was low. However, we noticed that the precipitates had a bright purple color (Appendix A). Moreover, the active protein could be transferred to a solution, solubilizing the precipitate by bicelles (we tested DMPC:F-EM). This means that protein was expressed in the active form, but was precipitated because the detergent failed to form bicelles under those conditions. On the other hand, we showed that DMPC or detergents used separately do not lead to the production of the active protein in solution, although a slightly purple precipitate was observed when DMPC liposomes were used (Figure 2, Appendix A). We believe that Facade detergents play the role of helper/transporter. They can surround the hydrophobic parts of the protein during ribosome synthesis and this complex then fuses with the lipid bilayer, where the protein is beginning to adopt its native form. Thus, the combination of CF synthesis, Facades, and specific lipids could be used in synthetic biology or further biophysical studies to elucidate the laws in the basis of membrane protein-lipid insertion and co-translational folding [62,63,64,65,66].

To summarize, we report here Facade-based bicelles, especially Facade-EM, as one of the best membrane mimetics for cell-free synthesis of the ESR membrane protein. The influence of bicelle composition and detergent concentrations in reaction and feeding mixtures on the yield of active ESR was determined. These data can be used for the production of other membrane proteins or become a basis for the search for other prospective membrane mimetics for cell-free systems.

## 4. Materials and Methods

### 4.1. Cell-Free Synthesis and Sample Preparation

A standard cell-free reaction was carried out in homemade reactors similar to those described earlier [67] using the 14 kDa (#D0405, Merck, Darmstadt, Germany) dialysis tube. The FM:RM ratio was 15:1 (50 µL of RM and 750 µL of FM). The reaction contained 100 mM HEPES (#H4034, Merck, Darmstadt, Germany) at pH 8.0, 0.83 mM EDTA (#E9884, Merck, Darmstadt, Germany), 0.1 mg/mL folinic acid (#47612, Merck, Darmstadt, Germany), 20 mM acetyl phosphate (#A0262, Merck, Darmstadt, Germany), 1.2 mM ATP (#1191, Merck, Darmstadt, Germany) and 0.8 mM each of GTP (#51120, Merck, Darmstadt, Germany)/CTP (#30320, Merck, Darmstadt, Germany)/UTP (#94370, Merck, Darmstadt, Germany), 2 mM 1,4-dithiothreitol (#D9779, Merck, Darmstadt, Germany), 0.05% sodium azide (#31803515, Molekula, Munich, Germany), 2% PEG-8000 (#89510, Merck, Darmstadt, Germany), 20 mM magnesium acetate (#M5661, Merck, Darmstadt, Germany), 270 mM potassium acetate (#1131, Gerbu, Heidelberg, Germany), 60 mM creatine phosphate (#27920, Merck, Darmstadt, Germany), 1 mM each of 20 amino acids, 0.2 tablet of complete protease inhibitor (#43203100, Merck, Darmstadt, Germany) and 0.1 mM of all-trans retinal (#18449, Cayman Chemical, Ann Arbor, MI, USA). The RM mixture additionally contained 0.5 mg/mL tRNA of *E. coli* (#12699020, Merck, Darmstadt, Germany), 0.25 mg/mL creatine kinase from rabbit muscle (#10127566001, Merck, Darmstadt, Germany), 0.05 mg/mL T7 RNA polymerase, 0.1 U/μL Ribolock (#E00384, Thermo Scientific, Boston, MA, USA), 0.08 μg/μL plasmid DNA, and 30% homemade S30 CF extract from BL21(DE3) strain of *E. coli* [14]. Reactions were conducted overnight at 32 °C and 150 rpm in an Innova 44 R shaker (Eppendorf, Hamburg, Germany).

To test the effect of detergents (DDM (#850520, Avanti, AL, USA), CHAPS (#C316S, Anatrace, Toledo, OH, USA), DHPC (#850305P, Avanti, AL, USA), and all Facades (#850562P-1EA, Avanti, AL, USA) on the protein synthesis, the corresponding membrane mimetic was added to the RM at the final concentration of 1% (*w*/*v*). To test the effect of the bicelles, 15 mM of lipid (DMPC (#D514, Anatrace, Toledo, OH, USA), DMPG (#D614, Anatrace, Toledo, OH, USA), DLPC (#850335P, Avanti, AL, USA), DOPC (#850375P, Avanti, AL, USA), POPC (#850457P, Avanti, AL, USA), or a mixture) and 30 mM of detergent (if otherwise not specified) were used in RM. To prevent dilution, the corresponding detergent was added to the FM with a concentration equal to CMC (0.17 mM DDM, 6 mM CHAPS, 15 mM DHPC, 0.1 mM F-EM, and 0.5 mM for other Facades), if otherwise not specified. To test the effect of liposomes, 30 mM of DMPC after extrusion through a 100 nm membrane (#800309, Cytiva, Little Chalfont, UK) was added into RM. To test the lipid-protein nanodiscs, a final concentration of 50 µM (estimated by MSP) was used. Nanodiscs were prepared as described earlier [68]. Briefly, the MSP1D1 protein was added to the mixture of DMPC:CHAPS (1:1.5 mole/mole, 80:1 DMPC:MSP1D1) after one cycle of freezing/thawing of the bicelles in an ultrasound bath at 30 °C for 15 min. After incubation at 30 °C for 1 h with shaking in the presence of 20 mM Tris HCl, 8.0, 70 mM NaCl, 0.02% NaN3, and 0.1 mM EDTA, BioBeads SM2 resin (#152-3920, BioRad, Hercules, CA, USA) was added to the solution (15 mg per 1 mg of CHAPS) to remove the detergent. The mixture was incubated overnight at 30 °C at the rotor mixer. The resin was removed by filtration/centrifugation. Incubation with a new portion of BioBeads was repeated three more times for at least 3–4 h each. The detergent removal was controlled by ^1^H-NMR.

### 4.2. Quantification of ESR Yields

After the overnight reaction, the RM was transferred to a clear tube. The aliquot was taken for Western blot analysis and RM was centrifuged at 25,000× *g* at 30 °C for 15 min. The supernatant was transferred to a clear tube and an aliquot was taken for Western blot analysis. The supernatant was used to measure and calculate the protein activity.

To calculate the yield of active ESR, UV–Vis spectroscopy was used. UV–Vis spectra were collected using ultra-micro ultraviolet–visible spectrophotometer ND-100 (Hangzhou Miu Instruments, Hangzhou, China) and processed by UVSRender software, based on the adaptive iteratively reweighted penalized least-squares algorithm [69]. The supernatant of the RM without DNA of ESR, tRNA, creatine kinase, T7 RNA polymerase, and Ribolock was used to set a baseline.

To estimate total and soluble protein yields, Western blot analysis was used. Proteins were separated according to their molecular weight by SDS–PAGE and transferred to a Nitrocellulose membrane of regular size (Power Blotter Select Transfer Stacks, Invitrogen, PB3310) by a Power Blotter device (Thermo Scientific, Boston, MA, USA). For one blot, four 12% Tricine gels were simultaneously transferred. The equivalent of 0.1 µL of RM was loaded onto each line of gel. 25 ng of pure ESR, control sample (total proteins after CF reaction without membrane mimetic), and 0.1 µL of prestained molecular weight marker (RAV11, Biolabmix-PS-2250, Novosibirsk, Russia) were added to each gel. After the transfer, membranes were washed with pure water followed by blocking for 1 h at room temperature with 1% BSA, 0.05% Tween-20 in 1× PBS buffer, incubated with antibodies (#MA1-21315-HRP Thermo Scientific, 1:10,000 in 1% BSA, 0.05% Tween-20 in 1× PBS buffer, overnight, 4 °C), and washed three times with 0.05% Tween-20 in 1× PBS buffer for 15 min at room temperature. Protein bands were visualized using enhanced chemiluminescent substrate (#32132, Thermo Scientific, ECL Plus) and imaged after 15 min of blot activation with a ChemiDoc MP Imaging System (Bio-Rad) in a high-resolution blot application in accumulation mode of 100 images for 50 s with software automatic optimization of the exposure for intense bands for an image area of 11.0 × 8.2 sm (WxL). Band intensities were estimated by Image Lab 6.0.1 Software (Bio-Rad) using the last (49.9 s) image in the accumulation row and custom parameters of band detection: custom sensitivity 50, sensitivity 10, size scale 3, noise filter 0, and shoulder 50. The molecular mass of ESR (28,848 g/mol) and molar extinction coefficients E_280_ = 46,870 and E_534_ = 43,000 were used to calculate the protein yield [22].

## Figures and Tables

**Figure 1 ijms-24-14864-f001:**
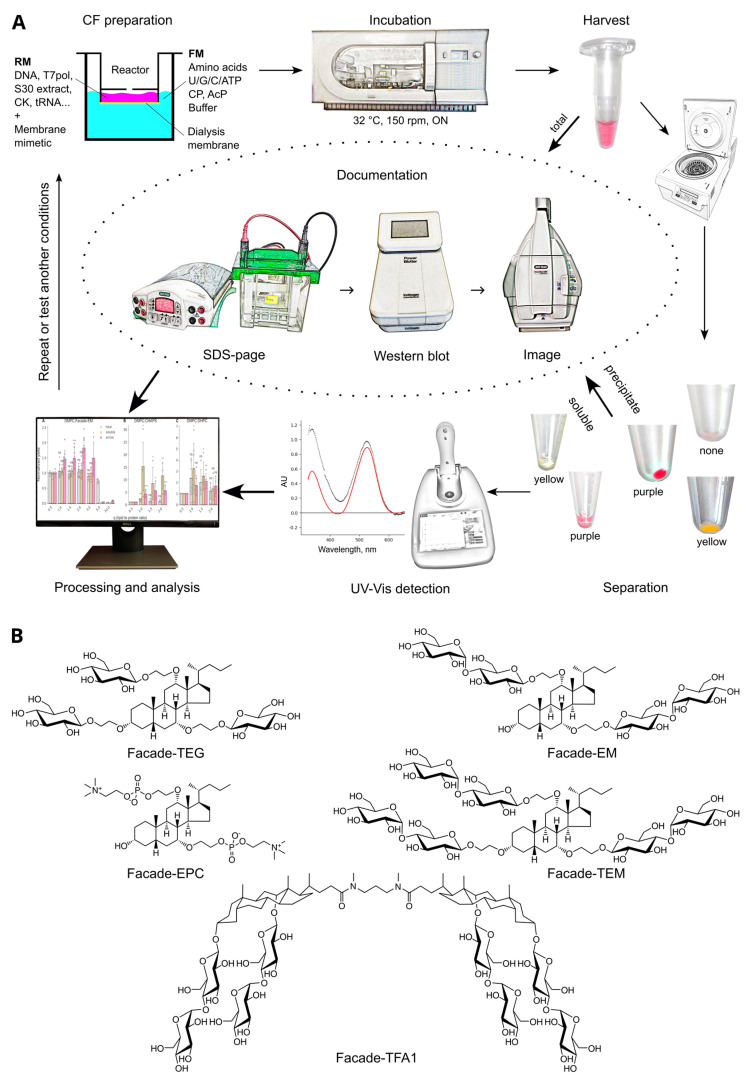
The scheme of the experiment (**A**) and the structures of Facade detergents (**B**).

**Figure 2 ijms-24-14864-f002:**
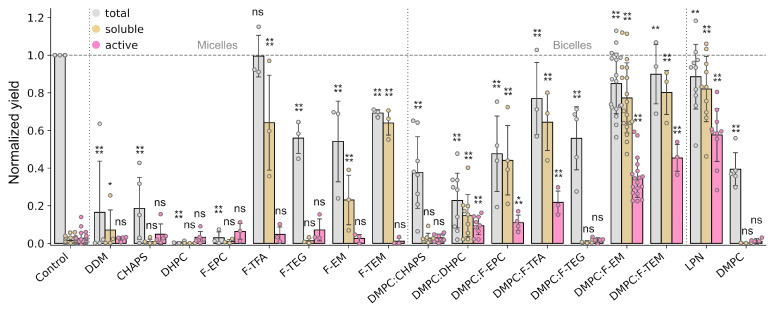
Comparison of cell-free expression levels of total, soluble, and active ESR in the presence of membrane mimetics. The total (gray) and soluble (yellow) expression levels were estimated by Western blot analysis. The yield of active (purple) ESR was quantified by UV–Vis spectroscopy. All yields were normalized to the total yield of the control sample (cell-free reaction without any membrane mimetics). For the micelles, the concentration of detergents in the reaction mixture was 1%. For the bicelles, 15 mM of DMPC and 30 mM of detergent, except F-TFA and F-TEM (15 mM), was added into the reaction mixture. For all reactions with detergents, a concentration equal to critical micelle concentration or higher was used in the feeding mixture to prevent detergent dilution. Error bars denote standard deviations; statistical significance is provided according to a multiple *t*-test with Holm–Bonferroni correction (*—*p* < 0.05, **—*p* < 0.01, ***—*p* < 0.001, ****—*p* < 0.0001, and ns denotes that changes are not significant). All statistical data are provided in Appendix A.

**Figure 3 ijms-24-14864-f003:**
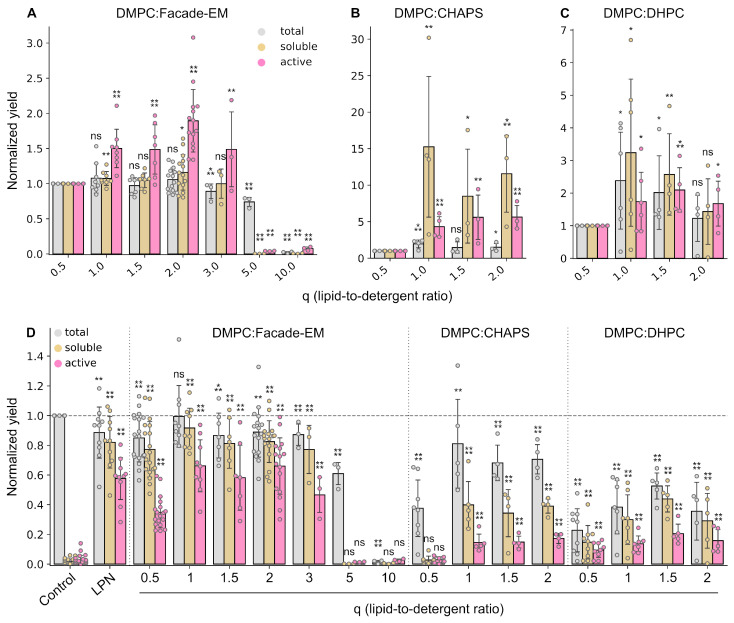
The influence of lipid-to-detergent ratio (q) on the CF synthesis of total, soluble, and active ESR for DMPC:Facade-EM (**A**), DMPC:CHAPS (**B**), and DMPC:DHPC (**C**) bicelles. The overall comparison of bicelles with normalization to the total yield of the control sample is presented (**D**). To evaluate the relative changes (**A**–**C**), the total, soluble, and active protein yields for all samples were normalized to the total, soluble, and active protein yields obtained for q = 0.5, respectively. The total (gray) and soluble (yellow) expression levels were estimated by Western blot analysis. The yield of active (purple) ESR was quantified by UV–Vis spectroscopy. 15 mM of DMPC was used for all samples in the reaction mixture. For each detergent, a critical micelle concentration was used in the feeding mixture to prevent detergent dilution. Error bars denote standard deviations; statistical significance is provided according to a multiple *t*-test with Holm–Bonferroni correction (*—*p* < 0.05, **—*p* < 0.01, ***—*p* < 0.001, ****—*p* < 0.0001, and ns denotes that changes are not significant). All the statistical data are provided in Appendix A.

**Figure 4 ijms-24-14864-f004:**
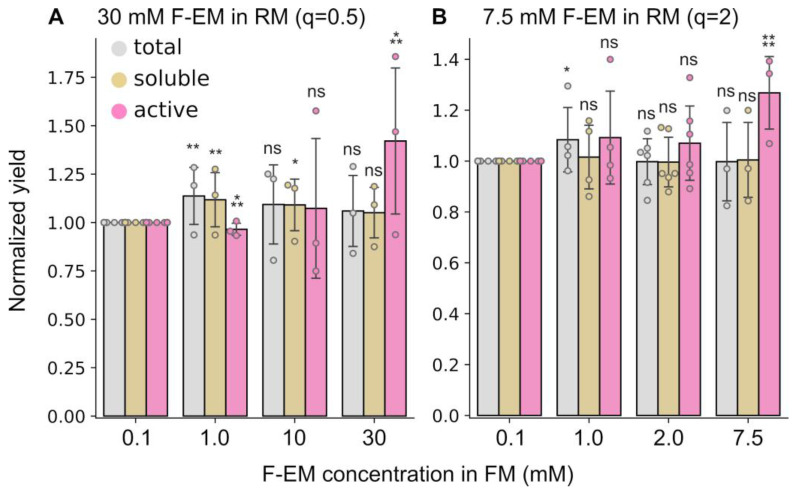
The influence of detergent concentration in feeding mixture on the CF expression levels of total, soluble, and active ESR for DMPC:Facade-EM bicelles. The detergent concentrations in RM were 30 mM (**A**) and 7.5 mM (**B**). To evaluate relative changes, the total, soluble, and active protein yields for all samples were normalized to the total, soluble, and active protein yields obtained for concentration of F-EM = 0.1 mM, respectively. The total (gray) and soluble (yellow) expression levels were estimated by Western blot analysis. The yield of active (purple) ESR was quantified by UV–Vis spectroscopy. Error bars denote standard deviations; statistical significance is provided according to a multiple *t*-test with Holm–Bonferroni correction (*—*p* < 0.05, **—*p* < 0.01, ***—*p* < 0.001, ****—*p* < 0.0001, and ns denotes that changes are not significant). All statistical data are presented in Appendix A.

**Figure 5 ijms-24-14864-f005:**
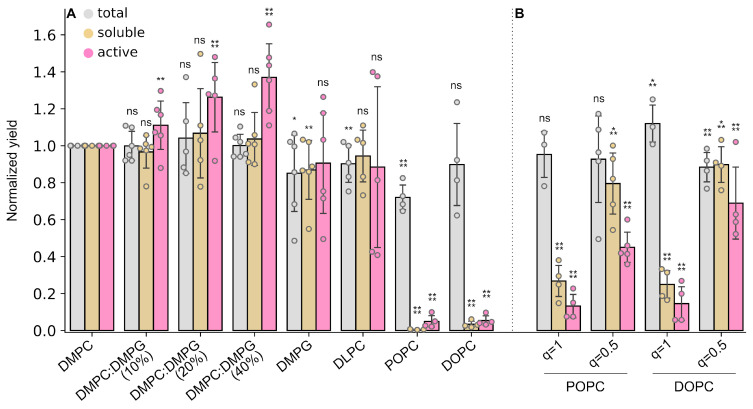
The influence of lipid composition of Facade-EM bicelles on the CF expression yields of total, soluble, and active ESR. (**A**) The effect of the lipid composition of Facade-EM bicelles prepared at q = 2 (detergent concentration in RM was 7.5 mM; in FM—0.1 mM). (**B**) The effect of q for POPC and DOPC bicelles on the protein yields. To evaluate relative changes, the total, soluble, and active protein yields for all samples were normalized to the total, soluble, and active protein yields obtained for DMPC:F-EM bicelles at q = 2, respectively. All the data normalized to control total yield are shown in Appendix A. The total (gray) and soluble (yellow) expression levels were estimated by Western blot analysis. The yield of active (purple) ESR was quantified by UV–Vis spectroscopy. Error bars denote standard deviations; statistical significance is provided according to a multiple *t*-test with Holm–Bonferroni correction (*—*p* < 0.05, **—*p* < 0.01, ***—*p* < 0.001, ****—*p* < 0.0001, and ns denotes that changes are not significant). All statistical data are presented in Appendix A.

**Figure 6 ijms-24-14864-f006:**
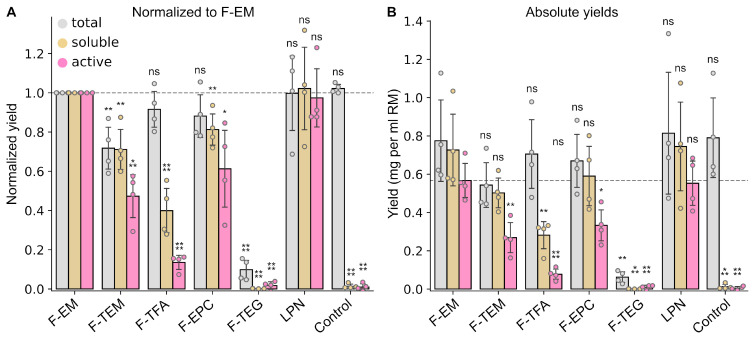
Comparison of ESR yields in the CF system in the presence of DMPC bicelles with different Facades. (**A**) To evaluate relative changes the total, soluble, and active protein yields for all samples were normalized to the total, soluble, and active protein yields obtained for DMPC:F-EM bicelles at q = 2, respectively. (**B**) The absolute yields of ESR (mg per ml of RM). The total (gray) and soluble (yellow) expression levels were estimated by Western blot analysis. The yield of active (purple) ESR was quantified by UV–Vis spectroscopy. Error bars denote standard deviations; statistical significance is provided according to the multiple *t*-test with Holm–Bonferroni correction (*—*p* < 0.05, **—*p* < 0.01, ***—*p* < 0.001, ****—*p* < 0.0001, and ns denotes that changes are not significant). All statistical data are presented in Appendix A.

## Data Availability

All experimental data used in the present study are available in the Appendix A.

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
