# Peer review of "Facade-Based Bicelles as a New Tool for Production of Active Membrane Proteins in a Cell-Free System"

_ijms, 2023, doi:10.3390/ijms241914864_

Round 1
Reviewer 1 Report
This manuscript summarizes data form an interesting and useful study that compares the use of a particular type of detergents, termed “facade”, as additives in cell free protein synthesis systems applied to produce membrane proteins. This methodology is highly relevant because the production of membrane proteins is a bottleneck in the study of their structure-function determinants. In this work, the authors have selected a given model protein, bacteriorhodopsin, and compared its production by a cell-free methodology in the absence and presence of different detergents, detergent/lipid mixtures or membrane-mimicking assemblies. Different operational factors have been also compared to show at what extent there is room for optimization in the different systems. As an additional interesting question, the authors have compared not only the yielding in “soluble” properly folded protein, but also its somehow functional potential, taking advantage of the intrinsic property of rhodopsin to exhibit unique optical properties when it is properly assembled.
In general terms, this work can be of great utility for the readers, and provides very interesting information as a starting point to optimize other different membrane protein expression systems. I only suggest completing the discussion of a couple of important questions to enhance the relevance and impact of the paper:
Obviously, everything determined and discussed in the text refers to the expression of bacteriorhodopsin. One wonders how different could be the optimal conditions to produce other types of membrane proteins. Would facade always be the most similar in terms of efficacy to LPNs? A bit of discussion in this respect, at least to let open the question of comparing other proteins, is warranted.
No doubt, one of the greatest advantages of facade detergents, once they have shown comparable efficacy to LPNs, is that they are a “open” system, allowing their complementation with many other factors with high versatility. It is a pity that the authors have not “played” a bit with their selected model to illustrate this potential, taking into consideration the unique character of bacteriorhodopsin as requiring a co-factor, the retinal molecule. As a matter of fact, it is surprising that retinal (crucial for bacteriorhodopsin purple colour) is not mentioned at all. When and where is it supplemented? The authors could have easily carried out experiments removing and supplementing retinal to show how their optimal systems allow a highly versatile manipulation. Please, details regarding the availability of retinal for the proper folding/reconstitution of the protein are strictly required.
I do not like the term “soluble” protein as it has been used in the whole text to refer protein that is properly assembled into membrane-mimicking environments that permit their stable suspension into aqueous systems. And particularly, I do not agree in that “membrane mimetics can solubilize the MP…”(lines 46-47 in page 2). I suggest instead stating that "membrane mimetics can assemble the MP…" At least, I would like to see a specific reference that when talking about MP solubilization, it is meant their assembly into membrane mimicking organizations small enough to form stable suspensions.
A list of abbreviations would be most useful for the readers.
English is fine.
Author Response
- Obviously, everything determined and discussed in the text refers to the expression of bacteriorhodopsin. One wonders how different could be the optimal conditions to produce other types of membrane proteins. Would facade always be the most similar in terms of efficacy to LPNs? A bit of discussion in this respect, at least to let open the question of comparing other proteins, is warranted.
- We agree that the situation with other membrane proteins could be different and included the discussion of this problem in the revised manuscript. Now we state: "On the other hand, we would like to point out that all the results here are based on the example of bacteriorhodopsin, which is a convenient model, but is known to be a well-behaving protein and cannot be considered representative of all the MP classes. Thus, while the effect of Facade detergents on translation machinery is independent on the type of synthesized MP, their ability to support the protein folding and stability needs to be verified in every particular case, because various proteins are known to require the individual membrane-like environment [16,23,61]".
- No doubt, one of the greatest advantages of facade detergents, once they have shown comparable efficacy to LPNs, is that they are a “open” system, allowing their complementation with many other factors with high versatility. It is a pity that the authors have not “played” a bit with their selected model to illustrate this potential, taking into consideration the unique character of bacteriorhodopsin as requiring a co-factor, the retinal molecule. As a matter of fact, it is surprising that retinal (crucial for bacteriorhodopsin purple colour) is not mentioned at all. When and where is it supplemented? The authors could have easily carried out experiments removing and supplementing retinal to show how their optimal systems allow a highly versatile manipulation. Please, details regarding the availability of retinal for the proper folding/reconstitution of the protein are strictly required.
- We thank the reviewer for this important finding and we have added the information about retinal to the text in the Results and Methods section. Regarding the manipulations with the retinal and bicelles, the retinal is partially soluble in water, thus it is available for interaction with rhodopsin in any of the media. Moreover, the general protocol for bacteriorhodopsin refolding is to transfer the protein into the bicelles in the presence of the retinal in solution (10.1006/jmbi.2001.4603). Thus, in this article, we did not conduct manipulations to study retinal properties but concentrated on protein expression depending on the membrane mimetics.
- I do not like the term “soluble” protein as it has been used in the whole text to refer protein that is properly assembled into membrane-mimicking environments that permit their stable suspension into aqueous systems. And particularly, I do not agree in that “membrane mimetics can solubilize the MP…”(lines 46-47 in page 2). I suggest instead stating that "membrane mimetics can assemble the MP…" At least, I would like to see a specific reference that when talking about MP solubilization, it is meant their assembly into membrane mimicking organizations small enough to form stable suspensions.
- We modified the sentence, specified by the reviewer. Now we state that "membrane mimetics that can support the native folding of an MP, self-assemble into the particles small enough to form stable suspensions". We would also like to point out that 'soluble protein' is not equivalent to the properly folded protein and the assembly into the membrane mimetic does not ensure the native folding. Properly folded protein corresponds to the "active" protein in the terms of the current study. We also defined the terms "soluble" and "active" protein at the beginning of the results section: "Here and below, 'soluble protein' stands for the protein that remains in solution after centrifugation, most likely embedded into the particles of membrane mimetic environment, but may not be active, whereas 'active protein' means a soluble and properly folded ESR."
- A list of abbreviations would be most useful for the readers.
- We added the list of abbreviations after the Methods section.

Reviewer 2 Report
The authors describe the evaluation of a new class of detergents for their usage in cell-free membrane protein production. They focus on the formation of bicelles by supplying the facade detergents in combination with lipids and they use bacteriorhodopsin as model membrane protein.
As the Facade detergents are emerging compounds with distinct properties and characteristics, a thorough evaluation of their suitability in cell-free systems is generally of interest and they may further expand the toolbox for membrane protein production. Unfortunately, the value of the presented evaluation is limited as only effects on bacteriorhodopsin production were studied. The extremely well-behaving bacteriorhodopsins are an exceptional case and they are not representative for the vast majority of complex and difficult membrane proteins. This should clearly be pointed out. Furthermore, some statements appear to be quite biased and the results need to be better discussed in context with previous literature.
- Please make clear in abstract or introduction that you work with an E. coli cell-free system. Also indicate the strain used for lysate preparation in the methods section.
- Line 50ff and discussion 279ff. The mentioned drawbacks of nanodiscs are not true. Nanodiscs are indeed very stable and can be stored even at 4oC for long time. Furthermore, expenses can be largely reduced by using home-made discs. In contrast, Facade detergents appear to be quite expensive as stated in line 307ff. Nanodiscs are also not static and rapidly exchange lipids and other materials which has also been published before. Cell-free synthesis of large membrane protein complexes in nanodiscs is furthermore possible and has been published e.g. in Köck et al 2022.
- Line 70/71: ..best membrane mimetic for MPs.., as it was only shown to be useful for bacteriorhodopsin, this is quite a bit exaggerated.
- DHPC, please indicate that you used hexanoyl (DH6PC) and not heptanoyl (DH7PC), as they show significant different characteristics in cell-free systems.
- Fig. 1: Please include a structure of Facade detergents as example
- Figures: Please generally give a value for normalized “1” in mg/mL
- Section 2.1: DPC, Chaps and DDM are not well tolerated in E. coli lysates. Their toxic effects were already described before, compare results with Klammt et al 2005. For DDM, 1% is far too high as already 0.1% results into significant synthesis reduction.
- Line 100: What is significant yield of ESR? Please give numbers and compare with other published data on bacterio- or proteorhodopsin expression.
- LPNs were formed with which lipid?
- L285: Include Smalps in discussion
- Line 296: Nanodiscs can be exchanged into isotropic bicelles as well (Laguerre et al 2016)
- L314ff: It is published that other rhodopsins form red precipitate as well in presence of liposomes in cell-free systems. To propagate Façade detergents even as translocons goes too far. The supplied lipid simply forms large liposomes by fusion which then precipitate in the RM and contain integrated bacteriorhodopsin.
Author Response
First, we are very grateful to the reviewer for his/her expertise and opinions. We received several very constructive comments that, as we hope, will improve our manuscript. To address the comments, we modified the text and figures of the manuscript. We highlighted all the substantial changes in the text in cyan color. Below we provide a point-by-point response to all of the comments.
- As the Facade detergents are emerging compounds with distinct properties and characteristics, a thorough evaluation of their suitability in cell-free systems is generally of interest and they may further expand the toolbox for membrane protein production. Unfortunately, the value of the presented evaluation is limited as only effects on bacteriorhodopsin production were studied. The extremely well-behaving bacteriorhodopsins are an exceptional case and they are not representative for the vast majority of complex and difficult membrane proteins. This should clearly be pointed out. Furthermore, some statements appear to be quite biased and the results need to be better discussed in context with previous literature.
- We agree with the reviewer that our study is limited to the behavior of a single protein and we rewrote the discussion of the revised manuscript to clearly point this out. Now we state: "On the other hand, we would like to point out that all the results here are based on the example of bacteriorhodopsin, which is a convenient model, but is known to be a well-behaving protein and cannot be considered representative of all the MP classes. Thus, while the effect of Facade detergents on translation machinery is independent on the type of synthesized MP, their ability to support the protein folding and stability needs to be verified in every particular case, because various proteins are known to require the individual membrane-like environment [16,23,61]."
- Please make clear in abstract or introduction that you work with an E. coli cell-free system. Also indicate the strain used for lysate preparation in the methods section.
- We have added clarification to the Introduction, Results, and Methods sections.
- Line 50ff and discussion 279ff. The mentioned drawbacks of nanodiscs are not true. Nanodiscs are indeed very stable and can be stored even at 4oC for long time. Furthermore, expenses can be largely reduced by using home-made discs. In contrast, Facade detergents appear to be quite expensive as stated in line 307ff. Nanodiscs are also not static and rapidly exchange lipids and other materials which has also been published before. Cell-free synthesis of large membrane protein complexes in nanodiscs is furthermore possible and has been published e.g. in Köck et al 2022.
- We would like to point out that according to biophysical studies, the exchange of lipids between the nanodiscs is substantially hindered. The exchange rate is comparable to the one observed in SUVs and is 3-4 orders of magnitude slower than the lipid exchange in micelles and SMALPs. The exchange in LPNs is driven by the monomeric lipids present in solution (https://www.nature.com/articles/srep45875), therefore the transfer of less soluble species, like membrane proteins or cholesterol is much less probable. Indeed, one can co-translationally assemble the dimeric membrane protein in LPNs, however mixing together samples of two different proteins in LPNs to obtain the functional complex is hardly possible and this is definitely a limitation. The stability of LPNs is also questionable, they cannot undergo freezing and once destroyed do not self-assemble in the absence of detergent. Finally, 200 mg of Facade-EM costs 710$ without any discount (https://avantilipids.com/product/850522) and we doubt that producing 200 mg of MSP could be cheaper, taking into account the costs for the growth medium, reagents and the labor hours. Moreover, for an adequate comparison, the cost of a homemade Facade must be estimated, which probably will be noticeably lower than the commercial version. Nonetheless, we do not want to state that LPNs are "bad" and bicelles are "good". Both mimetics have their advantages and drawbacks and it is proficient to have a wide variety of environments instead of the single available option. We rewrote the introduction and discussion of the revised manuscript to list both the advantages of drawbacks of LPNs taking into account this comment of the reviewer. We now state in the introduction: "The lipid-protein nanodiscs (LPNs) have shown a significant superiority over other mimetics [22], however, they have several drawbacks, including relatively high cost, which can be reduced by the in-home synthesis of belt proteins, and the hindered exchange of matter between the particles [27]. While there are examples, with protein complexes being co-translationally incorporated into the LPNs [28], further possible manipulations with the protein samples obtained in LPNs are limited.".
- Line 70/71: ..best membrane mimetic for MPs.., as it was only shown to be useful for bacteriorhodopsin, this is quite a bit exaggerated.
- We changed to "prospective membrane mimetic".
- DHPC, please indicate that you used hexanoyl (DH6PC) and not heptanoyl (DH7PC), as they show significant different characteristics in cell-free systems.
- This is actually explicitly stated at the beginning of Section 2.1, when the DHPC abbreviation was first introduced. We also added the abbreviation section to the manuscript after the Methods section.
- Fig. 1: Please include a structure of Facade detergents as example
- We added Facade detergents to Figure 1.
- Figures: Please generally give a value for normalized “1” in mg/mL
- We provide the final absolute yields on the final Figure 6 of the manuscript, and they are about 0.8 mg per mL of reaction mixture. Since in different figures, we normalize to different values to have a clear dependence, we added a sentence in Section 2.1, that states that the yield of protein in the absence of membrane mimetics corresponds to circa 0.8 mg/mL.
- Section 2.1: DPC, Chaps and DDM are not well tolerated in E. coli lysates. Their toxic effects were already described before, compare results with Klammt et al 2005. For DDM, 1% is far too high as already 0.1% results into significant synthesis reduction.
- We mentioned the fact that the toxic effects of these detergents were shown previously in the revised version of the manuscript.
- Line 100: What is significant yield of ESR? Please give numbers and compare with other published data on bacterio- or proteorhodopsin expression.
- We now specify that a significant yield of active protein means “starting from 30% compared to the total yield of the control sample”. Additionally, we compared the obtained yield of active ESR to the previously published data for this particular protein (Lyukmanova et al., 2012) in Section 2.5.
- LPNs were formed with which lipid?
- We used DMPC based LPNs. We included the LPNs preparation to Section 4.1.
- L285: Include Smalps in discussion
- We included SMALP in the discussion.
- Line 296: Nanodiscs can be exchanged into isotropic bicelles as well (Laguerre et al 2016)
- We agree with the reviewer and we mentioned this fact in the text “although this option is also available for LPNs through the stage of bicelles formation [59]”.
- L314ff: It is published that other rhodopsins form red precipitate as well in presence of liposomes in cell-free systems. To propagate Façade detergents even as translocons goes too far. The supplied lipid simply forms large liposomes by fusion which then precipitate in the RM and contain integrated bacteriorhodopsin.
- We agree with the reviewer that СF precipitates obtained in the presence of liposomes had a purple color, although less intense compared to bicelles (please see Supplementary Figure 3D). We also agree that the idea of “Façade detergents as translocons” is speculative. We modified the text and now we state “On the other hand, we showed that DMPC or detergents taken separately do not lead to the production of the active protein in solution, although a slightly purple precipitate was observed when DMPC liposomes were used (Figure 2, Supplementary Figure 3D). We believe that Facade detergents play the role of helper/transporter. It can surround the hydrophobic parts of the protein during ribosome synthesis and this complex then fuses with the lipid bilayer, where the protein is beginning to adopt its native form.”.

Reviewer 3 Report
The manuscript by Goncharuk et al. aims to explore the applicability of façade-based bicelles in the cell-free synthesis of membrane protein. The experiments and data analysis were well-designed and thorough, providing many details about optimizing membrane protein yield. I appreciate that the results are presented concisely and logically. I believe that the results presented here are worthy of IJMS. I only have some minor comments:
Some specific questions:
1. Line 131: “Besides, the size, shape, and phase transitions in F-EM-based bicelles are well established, unlike the other compounds.” The sizes of bicelles corresponding to different q values should be given if available. Also, the MSP used in the study and the nanodisc diameter(s) should be specified. These numbers will help the readers understand the Discussion here (Line 299): “As is known, the low q values correspond… and its partial unfolding.”
2. Line 202: “Surprisingly, this effect does not depend on the…. the optimum of q was equal to 1.” This is a very interesting result. Do the authors know if the DMPG percentages in the bicelle vs. in the RM are the same? My concern is that the DMPC:DMPG bicelle is not always a homogeneous mixture when changing the percentage. It is possible that ESP protein preferably interacts with the negatively charged lipids so that only the few DMPC molecules surrounding the ESP play a role in facilitating protein production. If this is true, it explains why DMPG in the 10-100% range can have a very similar effect. The ESP protein may only need a few DMPG molecules after all.
3. In the Discussion, the authors claim that (Line 280) “First, the LPNs do not exchange the components. This greatly complicates the study of protein-protein interactions in the membrane.” I am not sure this statement is accurate. Actually, the protein-protein interactions can be studied when the target MPs are incorporated into the same nanodisc (For example, https://doi.org/10.1073/pnas.2220477120). I would suggest the authors rephrase the sentence and be more specific about ‘do not exchange the components’ and add more references here.
4. In Figure S1, the DOPC molecular structure is mislabeled as DOPG.
Author Response
First, we are very grateful to the reviewer for his/her expertise and opinions. We received several very constructive comments that, as we hope, will improve our manuscript. To address the comments, we modified the text and figures of the manuscript. We highlighted all the substantial changes in the text in cyan color. Below we provide a point-by-point response to all of the comments.
- Line 131: “Besides, the size, shape, and phase transitions in F-EM-based bicelles are well established, unlike the other compounds.” The sizes of bicelles corresponding to different q values should be given if available. Also, the MSP used in the study and the nanodis
- We added the bicelle radii for different q values to the text and described the LPNs preparation in Section 4.1.
- Line 202: “Surprisingly, this effect does not depend on the…. the optimum of q was equal to 1.” This is a very interesting result. Do the authors know if the DMPG percentages in the bicelle vs. in the RM are the same? My concern is that the DMPC:DMPG bicelle is not always a homogeneous mixture when changing the percentage. It is possible that ESP protein preferably interacts with the negatively charged lipids so that only the few DMPC molecules surrounding the ESP play a role in facilitating protein production. If this is true, it explains why DMPG in the 10-100% range can have a very similar effect. The ESP protein may only need a few DMPG molecules after all.
- The empty PC:PG bicelles form a homogeneous mixture (10.1021/acs.langmuir.8b01454), but we agree with the reviewer that ESR can change the balance. We added one possible explanation for the dose-independent effect: “The DMPG dose-independent effect can be explained by the specific binding of the lipid to the ESR, for example, if the protein binds only a few molecules of charged lipid with high affinity and thus stabilizes the structure.”.
- In the Discussion, the authors claim that (Line 280) “First, the LPNs do not exchange the components. This greatly complicates the study of protein-protein interactions in the membrane.” I am not sure this statement is accurate. Actually, the protein-protein interactions can be studied when the target MPs are incorporated into the same nanodisc (For example, https://doi.org/10.1073/pnas.2220477120). I would suggest the authors rephrase the sentence and be more specific about ‘do not exchange the components’ and add more references here.
- We agree with the reviewer and have rephrased the sentence to clarify our point. Now we state “First, the LPNs almost do not exchange the components. The exchange of lipids between LPNs has rates, comparable to the observed in small lipid vesicles and three-four orders of magnitude lower than the ones in micelles or lipodiscs, and most likely follow the monomer diffusion mechanism [27]. While one can co-translationally assemble relatively large protein complexes in LPNs [28], mixing of proteins from two different LPN samples is hardly possible. This could complicate the study of protein-protein interactions in the membrane.”.
- In Figure S1, the DOPC molecular structure is mislabeled as DOPG.
- Corrected.
